# Three-Pillar Sustainability and Brand Image: A Qualitative Investigation in Thailand's Household Durables Industry

**Preechaya Chavalittumrong** [1,2,3] **and Mark Speece** [1,*]

1  Center for Research on Sustainable Leadership, College of Management, Mahidol University, Bangkok 10400, Thailand
2  College of Innovation, Thammasat University, Bangkok 10200, Thailand
3  10DK (Interior and Architectural Design), Bangkok 10110, Thailand
*  Correspondence: markwilliam.spe@mahidol.ac.th

**Abstract:** Many companies nowadays implement sustainable practices internally, and build brand images that communicate sustainability. However, there are different degrees of 'sustainability'. This study examines the extent to which full three-pillar sustainability (environmental, social, economic) translates into a sustainable brand image among consumers in Thailand. Nine major companies producing household durables were scored based on their website information, using the Dow Jones Sustainability Index to identify those having high-, mid-, and low-level sustainability implementation. In-depth interviews were conducted with three managers in one company at each level, and three consumers who mainly buy household durables from each company were also interviewed. Manager interviews confirmed that the level of sustainability implementation evident on the website is fairly accurate. Consumers roughly translate this into brand image reflecting the degree of the company's sustainability, but the mapping is not exact. Stronger communications about the company's sustainability seem able to improve consumer perceptions somewhat. Consumers are quite aware of three-pillar sustainability, but often do not explicitly consider all three pillars in their product decisions. However, the long-term trend seems to be toward merging the separate market segments into a comprehensive, three-pillar sustainability-oriented segment.

**Keywords:** sustainability; brand image; corporate image; sustainable production; corporate culture; corporate social responsibility; green marketing; Thailand

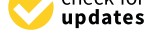



## 1. Introduction

Sustainability has become a critical issue in the modern world. A great many companies now implement sustainable practices internally, and externally aim to build a brand image that communicates sustainability. To date, however, progress toward incorporating the full concept of sustainability is somewhat lacking in the literature, particularly in marketing. Sustainability is built around three pillars, namely social, environmental, and economic elements [1,2]. Scholarly research on brand image rarely addresses more than one pillar at a time. There is little work on whether most companies actually fully implement all three pillars internally, or whether this translates into three-pillar brand images externally, and even some sustainability indices may lack full coverage. This qualitative study examines representative companies in Thailand's household durables industry to assess the extent to which their internal operations and external brand image are consistent with the S&P Corporate Sustainability Assessment (originally the Dow Jones Sustainability Index, DJSI), one of the few indices which do cover full three-pillar sustainability [3,4].

This is an important issue because focus on only some sustainability elements may not fully capture the benefits of sustainability, either in terms of broader impact from the company's internal operations, or in building an attractive brand image externally for increasingly demanding customers. The United Nations has laid out seventeen sustainable

development goals [5] which most three-pillar discussions cite. Although originally a separate schema, Dalampira and Nastis [6] demonstrate that these UN SDGs and the three pillars schema are essentially equivalent. "The 17 SDGs are integrated—that is, they recognize that action in one area will affect outcomes in others, and that development must balance social, economic and environmental sustainability" [7]. Action on individual sustainability elements or an individual pillar is certainly necessary, but not sufficient for reaching sustainability goals.

Most discussions of 'sustainability' in marketing focus on either green or corporate social responsibility (CSR) issues. For example, although they mention that cleaner production can consider economic, social, and environmental benefits of an organization's activities, de Oliveira Santos et al. [8] nevertheless focus on the environmental impacts of service production. CSR discussions also sometimes mention three-pillar sustainability, but do not really address all three, often focusing on the CSR regarding the environmental pillar (e.g., [9]) or the social aspects of CSR's stakeholder interests, such as in HR policy (e.g., [10]). Similarly, research on consumer response is somewhat fragmented. Armstrong et al. [11], for example, report positive perceptions of product–service systems for clothing to reduce pressure on the environment, although there is also some skepticism that companies actually do everything they claim. Ferrell et al. [12] demonstrate that consumers respond positively to strong CSR and corporate ethics.

Such work is important, but assessing all three pillars together is also essential. Discussing the context of financial services, for example, Ajour El Zein et al. explicitly conclude that more research is needed on "managerial implications on a practical level with an integrated model that takes into account the social, environmental and economic performance for the creation of sustainability-oriented brand value" [13] (p. 13). The interconnections noted above [7] can sometimes be mutually conflicting, and it may be necessary to strike a balance among them for optimal results (e.g., [14,15]). It is useful to ask whether the literature reflects a situation where companies are not yet thinking about full three-pillar sustainability, or whether current research is somewhat lagging behind company practice. Thus, the research questions examined here are as follows:

RQ1: How do companies implement sustainability in their internal operations? Do they address the full set of the three-pillar issues (as measured in the DJSI)? Can we identify different degrees of sustainability implementation?

RQ2: Can we assess this from publicly available information (websites), or do we need to get inside? (i.e., does public communication match what companies really do internally?)

RQ3: How do consumers perceive these companies? Do they think in terms of the three pillars, or do they perceive sustainability as one or maybe two pillars? Does this translate into a strong brand image of the companies' products?

There is not much empirical academic work examining how a full three-pillar conceptualization works in organizations, especially as it translates into brand image, and here, a qualitative approach was adopted. Qualitative research is most appropriate in the early stages of exploring most complex topics [16,17]. The assessment started with content analysis of corporate websites, and was followed up with in-depth interviews among corporate managers and consumers. The context of the investigation is Thailand, where there is a unique Thai sustainability framework called Sufficiency Economy Philosophy (SEP) [18,19]. SEP is essentially a version of Buddhist economics [20] that has been officially promoted for several decades [21]. Of course, it is not universally applied, and can be somewhat rudimentary even when it is. Nevertheless, some degree of sustainability consciousness has diffused widely in industry and among the general public, making Thailand a good place to examine these issues outside of Western contexts where they more commonly receive attention.

*Literature Review*

Sustainability is among the greatest challenges for companies in the modern world. Examining how to achieve sustainability, however, requires looking well beyond the

company itself, and understanding interconnections between internal and external factors. To the United Nations, "sustainable development is development that meets the needs of the present without compromising the ability of future generations to meet their own needs." [22] (Chapter 2 intro). That inability of future generations to meet their own needs could clearly bring major problems for everyone in the long term. The original UN definition, however, is somewhat vague, and there has been considerable work to define what this means. Current definitions provide more detail, either in terms of the UN's 17 Sustainable Development Goals [5,7] or the three-pillar framework [1,2], which are essentially quite similar [6].

The three-pillar concept is most often applied at the macro scale. There are frameworks to address these issues at the micro level, although mostly approached from an internal operations perspective, rather than marketing. The triple bottom line (3BL) is one of the most common, and often used as a management tool at a firm level [23]. The idea behind 3BL is that a firm's success should not be evaluated purely by the financial bottom line, but also by its social and environmental performances. One recent review [24] of research on sustainability issues in current industry trends (Industry 4.0), for example, indicates that much current discussion is about 3BL, which is comprised of (1) social sustainability, with emphases on social development and human capital; (2) environmental sustainability, which focuses on resource management that leaves the smallest carbon footprint; and (3) economic sustainability, which entails company profitability and liquidity. These three components are similar to the three-pillar concept, although the economic pillar in 3BL is somewhat oriented toward internal company economics, specifically toward profitability. Half or more of recent publications on Industry 4.0 in recent sustainability literature [24,25] are still conceptual rather than empirical investigations. Coverage is mostly internal operations, although both of these reviews briefly note the need for more work connecting internal operations and customers. Ultimately, though, there is no conflict between internal operations, customer orientation, and macro-scale sustainability, if individual business units operate with careful implementation.

The sustainability issues in Industry 4.0 discussion are not new. Society has been putting ever more emphasis on concern for the future of people and the environment, and businesses have thereby been challenged to integrate health, social, environmental, and safety concerns into their operations management. For some time now, this has focused substantial attention on sustainable operations systems [26], which in turn produce sustainable products and services. Sustainable products are items that provide social, environmental, and economic benefits concurrently. They are becoming a new standard in the market among customers. Much research shows that sustainable products and services are believed by consumers to be socially and environmentally responsible, which contributes significantly to customer satisfaction, purchase intention, and brand equity [11,27–30]. A strong brand image among consumers who value sustainability plays a key role in all this, although research that demonstrates it usually focuses mainly on green (e.g., [31,32]) or CSR [33] issues, rather than the three pillars together.

Probably the best way to support the sustainability message to the public is simply to integrate sustainability concepts into business practices and organizational culture. Corporate Social Responsibility (CSR) activities, when consumers are aware of them, normally convey such a message [34]. This can enhance the company's reputation and creditability in the rapidly expanding segments which care about such issues [35,36]. The use of CSR varies among organizations. Passive reactions to CSR issues can fail to get much impact, but when CSR is implemented proactively it can contribute substantially to brand and corporate image [37–40].

These brand and corporate image issues are one of the most common market reasons for a company to implement sustainability practices. (Companies can genuinely believe in sustainability, of course, but even so, they would likely want some market advantage from it.) Although a sustainable brand image can only attract customers who value sustainability (e.g., [41–44]), these market segments are rapidly expanding [45]. Companies aiming for

a long-term presence in the market need sustainability strategies to remain aligned with market trends [46]. Generally, stronger integration of sustainability in operations, products, and organizational culture, fosters a stronger sustainable brand image.

For example, green quality is often built in to attract consumers who are environmentally concerned [47,48]. Likewise, a social brand image is created to attract people who value socially responsible behavior [49]. A number of studies examine such green and social brand images and their relationship with other marketing components [50–53]. These studies, however, focus on one pillar at a time.

Thus, despite many studies on consumer perceptions of a specific pillar of sustainability, very few address full three-pillar sustainability. A recent bibliometric search of marketing journals in Scopus indicates that over three-fourths of articles published 2018–2021 cover only one pillar, usually green or social. Only about two percent examine all three pillars, and these proportions have changed very little compared to the period 2000–2017 (Figure 1; [54]).

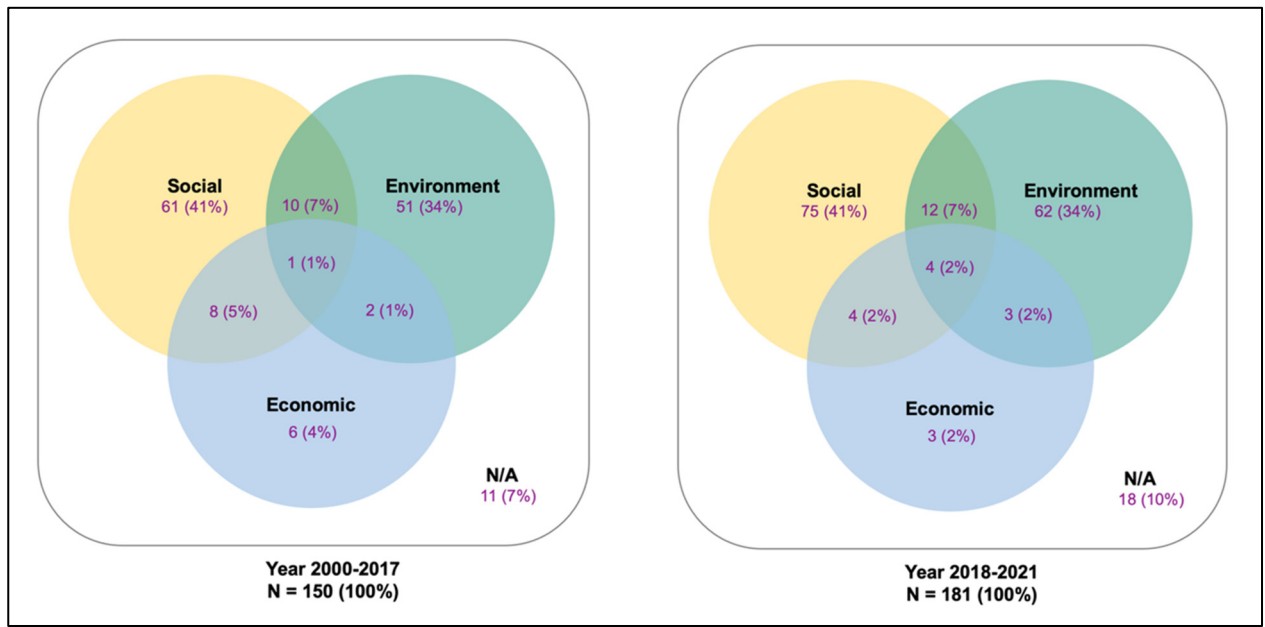

**Figure 1.** Sustainable branding research in SCOPUS marketing journals, 2000–2021. Source: authors.

Green practice leads to a green brand image, which is attractive primarily to green customers (e.g., [28,30–32,55]). Likewise, CSR and social marketing create a social brand image that attracts people who value social contribution and responsibility (e.g., [12,33]). There is evidence that some consumers are willing to pay a premium for the brands and products that possess green and/or social attributes that they value [56,57]. However, sustainability segments are growing rapidly, and they are merging as consumers become more knowledgeable about comprehensive sustainability and the interconnections among pillars. A one-pillar brand image may have worked so far, but long term, a stronger three-pillar brand image is likely to become essential. Some observers have already begun to point out that research on the simultaneous contribution of all three sustainability pillars to brand image is needed [13].

## 2. Materials and Methods

Because there is not much work examining how a full three-pillar conceptualization works in an organization, an exploratory approach was adopted, starting with content analysis of corporate websites, and followed up with in-depth interviews [16,17]. Household durables companies that sell consumer brands were chosen because the industry is fairly strong in Thailand [58], and these are high-involvement products. Managers may consider

sustainability issues with any kind of product, but here we aim to connect internal elements with external consumer perceptions. Consumers tend to build their preference and purchase intention with regard to these high-involvement products by actively gathering and assessing information [59,60]. Such high-involvement products are often considered more suitable when the researcher wants to examine more detailed thinking about how sustainability affects consumer perspectives (e.g., [61,62]).

*2.1. Choosing Companies*

The following five screening criteria were used to identify companies for initial assessment and potential inclusion in the qualitative interviews. (1) They must be local Thai organizations, so that their policies are determined locally, not set by corporate headquarters somewhere outside Thailand. (2) They are large-sized companies (annual revenue ≥ THB 500 million) registered with the Thailand Department of Business Development. Most such companies have a corporate website with information that can be assessed. Large companies' products are visible in the market, and there is a large base of consumers familiar with the brands. (3) The firms produce and sell household durables, such as home furnishings, homebuilding supplies, household appliances, and housewares. (4) The companies had to have a large presence in B2C (business-to-customers) markets, so that we could assess the impact of sustainability issues on products consumers actually buy, not just general impressions of products about which consumers rarely make choices. (5) They must integrate sustainability into their business policies and have been consistently executing policies such as CSR activities, and/or offering green or social products. They did not need to be leaders in this; we wanted variation in sustainability levels among the selected companies so that we could assess whether/how much consumers noticed different degrees of sustainable operations. However, we did not aim to assess companies that are not doing anything about sustainability at all.

An initial online search was performed by using the keywords referring to household durables and sustainability issues in both English and Thai languages. Keywords included terms both for products (such as household durables, furniture, sanitary products, home decorations) and for sustainability (such as sustainability, green, environmentally friendly, social, CSR, circular economy). Nine local Thai companies met the criteria stated above. They were double-checked in databases of the Department of Business Development, the Ministry of Commerce, and the Stock Exchange of Thailand to be sure they qualified as large companies. Five of them are public companies listed on the Stock Exchange of Thailand (SET), and the others are privately held.

The SET list, in particular, demonstrates that the listed companies ranked 1, 2, 3, 5, and 6 among the nine top listed companies (by revenue) in the industry. The other four companies do not include sustainability issues on their websites, so were judged not to be particularly proactive in implementing aspects of the three pillars. One was mostly business-to-business (B2B), so would not fit our target anyway. Companies below the top nine would not normally be considered 'big' in Thai practice. (Numbers 8 and 9 were borderline in terms of whether most observers would consider them 'big'). The revenues of the four privately held companies were determined from the other lists, and compared to the top SET companies; they fit within the range of 'big' companies we targeted. It should be noted that a few other big companies do sell consumer durables, but this is a small part of their overall product line. We did not want to confuse the discussion of sustainability in the industry with aspects that might have been implemented because of the characteristics of a different industry.

The chosen corporations were then scored according to their level of sustainability integration and execution. The scoring criteria were adopted from the DJSI. A number of sustainability indices are available, but this one is widely used and incorporates elements from all three pillars of sustainability, including economic, environmental, and social dimensions. Each industry has its own criteria [3,4]; the 2021 criteria for household durables (DHP) were used here. The DJSI 2021 scoring criteria [3] for household durables are divided

into three dimensions, including (1) governance and economic dimension (weight = 50), (2) environmental dimension (weight = 22), and social dimension (weight = 28). Each dimension has a number of sub-topics to specifically indicate how well a company implements sustainability in a range of different tasks. It should be noted that the 2022 weights for DJIS have now been adjusted to be somewhat more equal across the three dimensions. However, as noted below, we originally checked whether weighting the three dimensions equally (which is now closer to the 2022 weights) would change the assessment of the companies. It changed the raw scores very little, and did not change company rankings at all.

The initial assessment started with the corporate websites of these nine companies, with the main focus on the annual report and any sustainability report the company might have. Each company was scored according to the DJSI criteria and weights. The score for each item ranges from 0 to 3, where

0 = the topic is not mentioned in the documentation;
1 = the topic is mentioned but there is no evidence of implementation;
2 = the topic is mentioned and there is evidence of implementation;
3 = the topic is mentioned, there is evidence of implementation, and evidence of a successful result.

Table 1 presents the results of this scoring for each pillar and the overall rank. The DJSI assigns different weights to each pillar, roughly based on how many indicators they have in the schema, but calculations using equal weights for each pillar give essentially the same score, and exactly the same ordering. The website/documents assessment yielded companies with a range of sustainability scores, with good dispersion. Essentially, RQ1 can be answered positively. At least as indicated by their published information, companies do implement sustainability across the range of pillars, but how thorough this is depends on the company, and somewhat on the specific pillar. There is a wide range of sustainability implementations. This initial assessment made it possible to identify approachable companies with high, middle, and low scores within the range.

**Table 1.** Sustainability scores assessed according to 2021 DJSI criteria and weights.

| Company | Econ | Environ | Social | W-Econ | W-Environ | W-Social | W-Sum | Rank | Equal Weights | Rank |
|---|---|---|---|---|---|---|---|---|---|---|
| **C1** | **100.0** | **95.5** | **85.7** | **50.0** | **21.0** | **24.0** | **95.01** | **1** | **93.73** | **1** |
| C8 | 84.0 | 95.5 | 100.0 | 42.0 | 21.0 | 28.0 | 91.01 | 2 | 93.17 | 2 |
| C3 | 76.0 | 81.8 | 92.9 | 38.0 | 18.0 | 26.0 | 82.01 | 3 | 83.57 | 3 |
| **C7** | **58.7** | **90.9** | **81.0** | **29.4** | **20.0** | **22.7** | **72.03** | **4** | **76.87** | **4** |
| C6 | 58.0 | 95.5 | 71.4 | 29.0 | 21.0 | 20.0 | 70.00 | 5 | 74.97 | 5 |
| **C5** | **52.0** | **45.5** | **65.5** | **26.0** | **10.0** | **18.3** | **54.35** | **6** | **54.33** | **6** |
| C4 | 64.0 | 31.8 | 50.0 | 32.0 | 7.0 | 14.0 | 53.00 | 7 | 48.60 | 7 |
| C2 | 46.7 | 54.5 | 39.3 | 23.4 | 12.0 | 11.0 | 46.34 | 8 | 46.83 | 8 |
| C9 | 54.0 | 47.0 | 32.1 | 27.0 | 10.3 | 9.0 | 46.33 | 9 | 44.37 | 9 |
| **weights** | 0.50 | 0.22 | 0.28 | | | | | | | |

Notes: w- indicates weighted scores. **Bold** companies are ones in the follow-up interview process.

## 2.2. Choosing Respondents

In-depth interviews were carried out in one company at each sustainability level—high, middle, and low (relative to scores among the companies examined). Given that the nine companies represent most of the target 'big' companies in the industry, there was no particular reason to worry much about which company to choose at each level. Thus, the three specific companies were chosen because the first author of the article, who has worked in this industry, has good connections inside them. In general, personal connections are often needed in international business research and such access is an important criterion in judgment sampling [17] (p. 52). Connections are essential to most research in Thailand's

relationship-oriented culture (e.g., [63]). "With its strong traditions of business secrecy . . . working through connections and introductions is frequently the only way to gain good access at any level of companies in Asia" [64] (p. 69).

Within each company, three managers were interviewed to cover a range of functions within the organization (Table 2), focusing on particular individuals who specialize in the topics or work in the field related to the issues [65,66]. This, of course, follows the standard practice for exploratory qualitative work: "In qualitative sampling, purposefully select participants who can best help you understand the central phenomenon that you are exploring" [67] (p. 77). The initial contact in each company was asked to direct us to managers who have substantial authority in the key functions most involved in sustainability issues; we wanted managers who have decision-making authority on implementing important aspects of sustainability. It is worth noting that research in Thailand indicates that managers in different functions, even within the same company, may have different priorities and differing opinions about what is important. However, they tend to be consistent in reporting what their organization actually does [68].

**Table 2.** Companies' sustainability levels, manager and customer respondents.

| Company's Sustainability Level | Manager Respondent | Manager Job Function | Customer Respondent | Customer Rating of Company's Sustainability Level |
|---|---|---|---|---|
| Company1 High | M1 | Product development | C1 | Medium |
| | M2 | General management | C2 | Medium |
| | M3 | Sustainability unit | C3 | High |
| Comany7 Medium | M4 | Product development | C4 | High |
| | M5 | Business development | C5 | High |
| | M6 | Environmental unit | C6 | High |
| Company5 Low | M7 | Sales and marketing | C7 | Medium |
| | M8 | Product development | C8 | Low |
| | M9 | Customer relations | C9 | Low |

Respondents were also asked to rate and briefly explain their companies' practices related to the 2021 DJSI scoring criteria. A semi-structured format was used, with all interviews using the same list of topics corresponding to the three-pillars issues covered in the literature. However, the flow of discussion followed what the respondents felt was most relevant, so the topics were not covered in the same order, or sometimes not in the same detail. The questions were open-ended, with probing when needed to encourage respondents to elaborate on their answers [16,17,67]. Each interview lasted approximately 40–60 min, and notes were taken, as well as a digital record. Responses were categorized into themes and sub-themes, and compared across respondents, including noting how extensive responses were at the different levels of company and how well they reflected the DJSI assessment [17,67].

The consumer sample needed respondents who were customers of the companies interviewed. It is not difficult to find consumer durables customers in general, but finding knowledgeable customers who buy from a specific company is slightly more difficult in industries with more than a few dominant players. The customer side included nine customer respondents, three who mainly bought products from each of the interviewed companies (Table 2). Snowball sampling was used, which works well for reaching participants who are somewhat hard to find [69,70]. The initial respondents—the seeds—were selected through the researchers' personal networks to ensure that they were knowledgeable, able to give required information, and willing to participate [71], i.e., again relying on purposeful sampling [67]. The need to accommodate Thailand's relationship-oriented culture [63] is also a factor in consumer research. Sadler et al. [72] note that snowball is useful for adapting sampling to specific cultural conditions, while Van Meter [73] demonstrates that carefully implemented snowball sampling can be quite representative.

Qualitative methodologists frequently cite sample sizes of from 3 to 10 as adequate in this sort of qualitative phenomenological study (e.g., [67] (p. 77)). The key issue is saturation—the point at which one gains little new information from additional interviews. The few researchers who have empirically assessed saturation found that nearly all of the key issues can be uncovered with as few as six interviews [74,75], although one may need more to get into substantial detail. Hardly anything new comes up, however, after about 12 to 14 interviews [74,75]. In our case, there was some difference among respondents representing the three companies in terms of how much detail they could discuss, particularly if a company was not doing much on a particular sustainability pillar. However, there was not much difference in the issues respondents brought up. Our focus on what the company does, rather than what the manager considers important [68], also facilitates saturation somewhat. Similarly, the general issues were not much different among consumers, although there was a range of focus, from mainly one or another of the pillars to all three.

## 3. Results

Operations management was among the business functions that started addressing sustainability issues quite early. The initial focus was somewhat oriented toward the environmental pillar, although other aspects of sustainability did gain attention even in early discussions [26]. More recent discussions explicitly include a three-pillar approach. "We define sustainable OM as the pursuit of social, economic and environmental objectives— the triple bottom line (TBL)—within operations of a specific firm and operational linkages that extend beyond the firm to include the supply chain and communities" [76] (p. 1). As noted above, however, this has not necessarily translated into the literature examining the impact of sustainability issues on brand image.

### 3.1. Operations in the Manager Interviews

The manager respondents in these interviews all implied that sustainable products cannot be created without operations management attention to sustainability. "Sustainable products are the products of sustainable operations" (M2). Understanding of exactly what this means varies according to the companies' sustainability scores. The environmental pillar was prominent in all the companies' discussions. Even in the high-score company, the discussion often starts with environmental concerns. "Sustainable operations here involve waste management, resource management, and pollution management" (M1).

Thinking about the environmental pillar, however, seems to have different levels of sophistication in different companies. In the low-scored company, the answers were often rather vague and focused on repeating a common standard that the respondents know the company follows. "In our factory we have to comply with several green standards such as ISO and E1" (M8). Thinking on this pillar has more depth in some companies, and is often explicitly tied to other pillars. M3, from the high-score company, discussed policies connecting waste to economic value and CSR.

"The main concept is that we need to transform the waste as much as possible. . . . The first implementation is called waste to value, where we modify our waste and donate it to whom who can make use of it. The second implementation is called waste to CSR, where we give and teach local community how to create value from this waste" (M3).

Circular economy concepts were very prominent in the high-score company. "Every practice is created based on the concept of circular economy, where we can pass the value of our products to the others" (M3). Applications of the circular economy (as opposed to conceptualizations) are not very thoroughly researched (e.g., [77,78]), but our qualitative interviews suggest that many Thai companies are familiar with and working on such issues. The circular economy implicitly ties environmental issues into the other sustainability pillars. M1 explicitly noted that one aspect of sustainability is "supporting the circular economy". M3 also explicitly extends waste issues to their impact on the communities where operations take place. "Sustainable operations is to produce zero waste from the operations. This waste needs to contribute to other parties and is harmless to

local communities" (M3). Pollution is also explicitly linked to the other pillars; "we need to concern about the pollution occurred during the production and its effect to the local community" (M1).

A factory may recycle its own waste to use as a raw material for other products: "the waste itself needs to be able to be reused or recycled. For example, the thread left from textile production will be transformed to other products" (M4). M3, quoted above, talked about teaching the local community how to create value from factory waste. Another respondent in the medium-scored company indicated that the company itself figures out how to use the waste from other companies. They work with food processing companies, for example, and their consumer durables factory "renewed the energy from burning macadamia peels and use it in the ceramic factory instead of choosing LPG" (M5). However, none of the low-scored company managers mentioned the circular economy concept.

The products resulting from operations were also discussed as "sustainable" by many managers. Many respondents focused on two factors for green products, which are materials and product performance. They suggested that green products should be made of eco-friendly or recycled material, which leaves the smallest footprint on the environment. "We use recycled/reused material for our products" (M3). Sustainable products in high- and mid-score companies can also extend their life cycle by passing on their value to the next owner.

"A sustainable product is one which can be reborn (recycled, reused). This product shall carry its value to the next owner, although the value can be decreased" (M1).

"I think sustainable products are the ones that can be reborn. They can continuously be used and circulated. In other words, they are immortal" (M4).

By contrast, managers in the low-score company often seem to perceive green products as simply a matter of using proper materials. "Most of sustainability integration is used for product development such as eco-friendly materials. We have various products that incorporate sustainability" (M7). One of their sustainable products is furniture made from substitutes for stone and wood. This is good for the environment because they "do not need to harm the rock mountain and natural resources" (M7).

The examples above, particularly in the high-score company, illustrate that when discussion gets into more detail, all three pillars become evident. For example, in listing issues relevant to implementing these several aspects of sustainable operations, M1 included "social concern, especially for local people", "genuine and long-term CSR initiatives". Managers in the mid-score company similarly recognize more than a single pillar: "sustainable products should not only satisfy the designer and end users, but also the environment and society" (M5). Such thinking, however, was not as evident in the low-scored company. There, the topic of continuing value to users mostly came back to greenness. "Our company integrated sustainability into the operations by developing several products made of green material" (M8). Sometimes safety was also noted: "sustainable products are the ones that are harmless to the environment and users" (M7).

"Harmless", or non-harm, is a prominent theme in most Buddhist societies (e.g., [79]), and fundamental in companies that take sustainability seriously. "We need to be concerned about the pollution which can occur during production and its effect to the local community" (M1). Economic thinking among most Buddhist reform movements in Thailand, however, goes well beyond non-harm, to "pro-active right livelihood", contributing to society [20,80]. This, of course, is essentially the top level of the corporate social responsibility (CSR) pyramid popularized by Carroll [81]. At this top level, "corporate social responsibility therefore builds on the basic economic and legal contracts between corporations and society, and tries to go beyond these to further the common good" [82] (p. 13). This would seem to be inherent in strongly sustainability-oriented companies, and was prominent in the high-score company here. It tends to create more sophisticated CSR thinking that focuses on long-term goals.

One respondent, for example, stated that genuine CSR should not just benefit the company and cannot be a one-time thing, but rather must benefit mainly society. She

said, "real CSR activities refer to the ones which also contribute to society, not only for our company's sake. We are responsible for helping them to achieve their long-term goals. To me, just donating money cannot be considered real CSR" (M3).

One of the activities M3 has engaged in is to educate local people to transform the waste from her factory into raw material. These products can then be sold back to the company. This way, mutual benefit between the community and company was created.

Finally, continuous improvement is prominent in the high-score company, whereas it was not mentioned by respondents working in the low-score firm. The high-score firm managers generally cited a wider range of things that they do, but were nevertheless concerned that they need to do more, and that there are still issues that must be addressed.

"We have green products, but not totally sustainable. We integrate the concept in the design and use recycled/reused material. But we have not yet made them be able to pass the value to the others well. ... Because we have not yet successfully offered comprehensively sustainable products, we focus a lot on our after-sale service because we want our customers to rectify the broken products instead of buying the new ones" (M3).

### 3.2. Policy and Corporate Culture

The literature reveals that top management leadership is crucial in initiating and executing sustainability in an organization [83,84]. All respondents agreed that the most effective way to initiate policies is to derive them from the CEOs and management's vision. When a CEO or top managers initiate a policy and act seriously, everything can be executed with less effort. For instance, "most policies came out from the CEO, who focuses mainly on sustainability and environment" (M6). "It is a top-down policy, everyone needs to follow. In my opinion, this is the most effective way to implement the concept of sustainability in an organization and integrate it in the operations" (M1). Of course, several things are needed for this to actually work well.

First, top management needs to truly understand the issues, and communicate them effectively. M3, from the high-scored company, explained that the policies given to her from the top managers were very concise and easy to understand. She explained two policies in some detail: (1) "waste to value" and (2) "waste to CSR". The "waste to value" policy, for example, is when a company aims to transform its waste into a raw material for new products either for themselves or other companies. Effective communication from top management is essential to convey the message throughout the organization. Some respondents believed that part of effective communication is that policies can be executed effectively by incorporating sustainability concepts into key performance indicators (KPIs). M5 from the company with medium sustainability level stated, "this is also implemented seriously through KPIs for every department". Such KPI policies were mentioned in both the high- and medium-scored companies, but not the low-scored one.

Second, top management involvement does not mean top management diktat. Bottom-up policy initiation is also common in companies with higher sustainability scores. Most employees have a mutual understanding of sustainability and its goals, and their input is welcomed. Such involvement from various levels of the company can be explicitly institutionalized into specific departments or committees, which often bring cross-functional expertise together. For example, "the parent company has formed a sustainability development board, which recruits its members from multiple business units" (M2). There may also be more specific teams, for example, an environmental unit. M3 in the high-scored company cited an example of this: "the Environmental Unit proposed [those] ideas for management's consideration". The medium-scored company also has an environmental unit that mainly focuses on environmental matters in the company. M3 also noted that "the latest policy theme is circular economy. They have this committee communicating everything about circular economy and try to execute things through various activities" (M3).

Policy initiation and communication in the low-scored company is a little bit different. The top management is involved, and respondent M8 mentioned that "our top managers have joined and mentioned about sustainability in the meeting and training, mostly about

the products, so that we can explain our customers". This was mostly telling employees what to say in describing products, not so much about getting employee input. M8 explicitly said "All policies are top-down, which is effective". The company with low sustainability concern lacks any particular unit working solely on sustainability: "no, we don't have one" (M7). This makes it more difficult to get input from employees at all levels.

Another possible way to initiate sustainable operations is to integrate it directly into the business model. This way, the daily business routine will automatically turn to sustainable operations and will be naturally communicated to employees without much additional effort. One respondent in the mid-level company said, "we hired locals and taught them various skills" (M5).

"The business model itself is very socially concerned. The goal of the organization is to create a job for the locals, as well as to educate and train them to be skilled workers. Therefore, the policies about social attributes are not only a top-down policies, but also an integration to our daily business routine" (M4).

The business goal of helping society by providing good jobs is not unusual in Thailand, and, for example, is common among owners and managers in Thailand's Buddhist reform movements (e.g., [80,85]). It is also prominent elsewhere among strongly spiritually-oriented small business owners [86]. The desire to serve other stakeholders, both within and outside the company, is characteristic of the servant leadership style in a wide range of contexts [87]. The leader fosters the organizational environment, "the culture that people in the organization have the same vision of sustainability and practice with such goals in mind" (M2).

The results from the company interviews indicate substantial variation in how well sustainability is integrated into the organizational culture. Organizational culture is the collective shared assumptions and behaviors in organizations (e.g., [88,89]), and as many of the quotes indicate, companies which score high on sustainability demonstrate strong shared understanding and behaviors specifically regarding sustainability issues. What they talk about usually broadly addresses sustainability issues, and is mostly directly related to how they structure operations. Sustainability and its components, such as CSR, are not separate, somewhat unrelated activities.

To a manager from the low-scored company, however, sustainable corporate culture is mainly making sure everyone is aware of the need to "focus on the impact of our product to the environment" (M7). In addition to understanding sustainability to mainly mean 'green'; they do have CSR activities, but these seem to be somewhat of a separate issue from operations. It may refer to helping the community in some way, including renovating a school in a rural area, planting trees, and various forms of donation. "Our company has several CSR activities which we can choose whether to participate, such as forest planting and donation activities" (M9). These activities are mostly optional, one-time activities. It is difficult to get long-term impact, either in terms of building corporate culture oriented toward three-pillar sustainability, or actually achieving long-term impact benefitting the local communities.

Overall, then, internal manager interviews clearly indicated that these companies implement sustainability to different degrees, as was initially implied by their scores on the DJSI assessed from website material (Table 1). After the interviews, DJSI was scored for each company again, this time based on interview content. Occasionally, an issue covered in DJSI indicators did not seem to be implemented quite as thoroughly as the website indicated. On other issues, the interviews suggested somewhat more than was evident from the website. However, scores on individual indicators were not radically different, and any raising or lowering of the score tended to roughly even out. The overall scores were not substantially different, and the slight changes were far too small to change the rankings on sustainability implementation (Table 3). RQ2 above can be answered positively—i.e., information publicly available on websites does seem to be sufficient to assess how extensively companies are implementing three-pillar sustainability.

**Table 3.** Comparing DJSI with re-scoring after interviews.

| | Based on Website | | From Interviews | |
| --- | --- | --- | --- | --- |
| Company | Weighted Sum | Equal Weights | Weighted Sum | Equal Weights |
| C1 (high) | 95.01 | 93.73 | 91.66 | 92.03 |
| C7 (mid) | 72.03 | 76.87 | 70.11 | 74.37 |
| C5 (low) | 54.35 | 54.33 | 54.35 | 54.33 |

*3.3. Brand Image in the Consumer View*

Good performance on any pillar can contribute to a favorable brand image among those consumers who value some aspect of sustainability [11,12,28,30,32,33]. However, to the extent that consumers perceive different degrees of sustainability discussed above, brand image should show some differentiation. Table 2 indicates the three respondents interviewed from each company. As already noted, the respondents did roughly follow the rankings based on websites and internal company interviews, although it is not a perfect mapping. There seems to be an important role for marketing and corporate communications in conveying the company's sustainability efforts.

The literature has occasionally demonstrated the need for good communications as well as actual internal implementation, so that consumers are aware of company efforts [90–92]. This issue was not a primary theme in our research, but the interviews contained hints of it consistent with the imperfect mapping from internal sustainability to brand image. In particular, the medium-scored company's marketing communications have highlighted their local social engagement for several decades. "I have also seen from TV that they help the locals since I was a child. So it is natural for me to assume that this brand is highly ethical and sustainable" (C4). This seems to have solidified consumer perceptions of sustainability related to the social pillar. The marketing communications of the high-scored company, which actually has stronger sustainability implementation, do point out sustainability issues, but tend to focus more on the products.

The product focus is somewhat effective at communicating sustainability. Consumer thinking about sustainability issues mostly seems to start with the product. It often focuses initially on the environmental pillar, although many (not all) consumers go well beyond only green issues. Most build their green brand image through the product attributes; for example, products made from recycled materials or products that consume fewer natural resources compared to competitors. One respondent who buys products from the top-scored company said, "I think of them as a green brand because their products consume much less water compared to other options available" (C3). Sometimes knowledge comes from media communications, "I have seen them (CSR activities) from social media and TV, but mostly social media" (C2).

Sometimes first impressions come through word-of-mouth; the respondent who buys products from the mid-ranked company and said she has seen its social involvement on TV since childhood, and implied that she partly built the green component of brand image by listening to her father, "My Dad told me that they are concerned about the environment" (C4). This initial focus on product greenness also helps the brand image of the low-scored company, which, according to the discussion above, mostly views sustainability as greenness, and offers some green products. One respondent who buys products made by the low-score company said:

"I wouldn't say they are a green brand, but yes, they do have a bit of green image to me. I think I built that green brand image through their products. I spent a lot of time in their showroom, and I found many of their products are made of eco-friendly materials" (C7).

Customers of the high- and medium-scored companies, however, go well beyond just thinking about the environmental pillar. As with the green pillar, they also build their social brand image around products, as well as the brand story and CSR initiatives. Some customers may build a social brand image when they know their money will go to help the local community, such as "The product itself is made by local tribes with local materials'

(C2). Many know the wider brand story. Of course, the story has been communicated that it is the social brand since the beginning" (C6). Awareness of CSR initiatives also contributes. "I saw they have done a lot of CSR activities, so it seems natural to me that their brand is related to social image" (C2).

For some respondents, the environmental and social pillars are intimately connected—they extrapolate that green must be socially responsible, or they may reason in the opposite direction: "I have seen them helping the local community and the environment so I assumed that their brand is green" (C6). Such recognition of interconnections among the pillars is especially important for respondents who value the economic brand image, which is less likely to be directly attached to the brand's products. For example, consumers might think that the brand has a sustainable economic image because the products are made by local people. "It is very obvious that they are a local brand that stimulates the economy in many ways" (C3). Some consumers explicitly consider all three pillars together, having created a three-pillar image from their experience with all angles of knowledge about and interaction with the brand.

"I have known this brand since I was a kid, and I saw them on TV doing a lot of activities that are good for local society, implying that they also help stimulating local economy. I also bought their green products. So I consider this brand sustainable". (C4).

A social (and sometimes economic) brand image, however, was only cited as important by customers who purchase the brands with high and medium sustainability scores. The low-scored company, which mainly views sustainability as the green pillar, tends to get similarly-minded customers who focus mostly on green brand image.

Brand image essentially connects the company's internal efforts toward sustainability to perceived advantages for customers. The respondents were all concerned about product quality to at least some degree, and strongly sustainability-oriented customers tend to link sustainability to product quality. They feel that a sustainable brand is likely to offer good quality products because the brand's and company's intentions are good. For example, one respondent who purchased from the high-scored company said that,

"Because they are concerned about sustainability, it implies that the product quality should be good. They care to invest in sustainability, so they are likely to invest in the product development as well" (C3).

On the other hand, customers who buy products from the low-scored brand do not necessarily see this linkage between sustainability and quality. "I think sustainability has nothing to do with product quality" (C8). These customers, however, are similarly concerned with product and service quality, even if they do not connect quality to the company's sustainability efforts. For example, "I think when I contacted their after-sales service or communicated with the salesperson, I can feel that they are quite ethical and care about customer" (C8).

Beyond actual product quality, a sustainable brand image itself can contribute to satisfaction among sustainability-oriented customers [37,93,94]. Many respondents referred to moral and emotional fulfillment, consistent with some literature demonstrating the emotional component of, e.g., green brand image [55]. (Note that genuine greenness is perceived positively, but greenwash has a negative impact on brand image [95]). Even customers of the low-scored company feel that purchasing sustainable brands somehow contributes to society. For example, respondent C7, who is mainly focused on green in thinking about sustainability, and does not see strong implications for product quality, nevertheless finds emotional value in buying the company's green products. "I feel like I can contribute to world sustainability by buying such a brand; it makes me feel like I am a good person". Similarly, C9 said, "I feel less guilty when spending a lot of money to consume things" and "I feel good to help the planet". Respondent C6, who views sustainability more broadly than just green, cites sustainable brand image as an important factor in satisfaction. "[It] depends on products. If it is special, not a basic need, it [sustainable brand image] is necessary because we buy satisfaction, and sustainability is one of my satisfactions".

Research indicates that sustainability is among the relevant purchase criteria for many consumers, although not always the most important (e.g., [96]). In our interviews, sustainable brand image, especially aspects related to emotional fulfillment, was frequently cited as a driver of purchase intention for those who value sustainability at medium and high levels. It contributes to some extent even for customers of the low-scored company. Sustainability "helps me make a purchase decision and I can pay even more to buy those brands" (C2). Respondent C5 mentioned that, "If the product is not cheap, I would consider it a splurge, so fulfilling my emotional needs is necessary". For customers who value sustainability somewhat less, such as C8, only green image matters because she primarily focuses on the green pillar. She said, "green image of the brand might lead to purchase decision, the others are optional".

The willingness to pay more, just noted, was common in the interviews. This is consistent with other research demonstrating that sustainability-oriented consumers are willing to pay more for products with a strong sustainability image, provided they are also perceived as high quality (e.g., [56]). All respondents are willing to pay a premium if the brand is associated with sustainability; how big a premium depends on the importance of sustainability to them. Even customer C9, who has relatively low sustainability concerns, and buys products from the low-scored company, is still willing to pay up to 10 percent more for sustainable products (mostly green in this perception). For respondents who value sustainability highly, the premium can be substantial, up to 50 percent more compared to unsustainable brands. Customer C6, for example, was quite explicit about the decision criteria and willingness-to-pay:

"When I purchase such a product, I weight product [characteristics] at 30 percent and 70 percent for the brand. I can actually go up to 50 percent more [in price] for sustainable brands if I can afford it".

How much sustainability do customers expect? These respondents all believe that sustainability is a required practice, and the discussion has already noted several times that even customers of the low-scored company expect some, usually greenness. Eight out of nine respondents implied that integrating all three pillars is preferable, but not mandatory. Most are willing to accept that one or two pillars is enough. For example, C8 said "Not completely necessary for three things. Only one makes me feel good and can be considered sustainable". However, one respondent with high sustainability awareness thinks it is important for a company to have three pillars for its brands to be considered sustainable. "To me, I think a brand needs to have all three pillars to be a sustainable brand" (C6). Many tend to view one or two pillars at present as an intermediate stage in moving toward more sustainability. Customer C3, who has high sustainability consciousness, explicitly suggested that one pillar at a time will naturally progress toward three-pillar sustainability eventually:

"Not really, I think only one is enough. Of course, it is good to incorporate three, but it is much more possible if each company is responsible for one pillar. Eventually, they will combine into three pillars in the larger scale anyway".

The final two quotes here, from C6 and C3, suggest the long-term trend. At the current stage of market evolution, nearly all of the respondents would like to see all three sustainability pillars, but most are willing to accept only one or two. C6 however, already wants three-pillar sustainability, and C3 believes that the individual pillars will eventually merge to include all three. Many companies already implement three-pillar sustainability, and once they are somewhat better at communicating that to the market, the majority of sustainability-oriented customers are likely to move toward C6's desire for a brand image including all three.

## 4. Discussion and Conclusions

A summary of the RQ that guided this work illustrates a key theme here: there is a need to focus more on three-pillar sustainability in scholarly research on sustainable brand image. This is similar to what several observers have recommended recently [13]. Much

research in marketing examines only a single sustainability pillar [54]. Our research enriches the discussion by demonstrating that many companies are implementing full three-pillar sustainability, but that they do so to different degrees. The companies with higher levels of sustainability, measured by DJSI criteria, tend to integrate all three pillars into their practice and develop a sustainable culture internally. For big companies at least, their degree of sustainability implementation can be roughly determined from public information on their websites (RQ1). There is a range of such implementations. Lower-scored companies are involved in sustainability, but not as extensively, and not comprehensively across all pillars. Nevertheless, if they are concerned with the issue, there seems to be some effort to cover the three pillars.

Internal discussion with managers confirms the relative degree of sustainability implementation (RQ2). It also uncovers additional detail, not least of which is that, in practice, companies with lower scores tend to focus on one or two pillars, particularly on green or CSR attributes (such as green with C3). The interviews demonstrated this, although the DJSI scores on individual pillars may not be entirely useful for distinguishing when a company focuses more on one or another pillar. The substantial interconnection between pillars can mean that a company focusing on green, for example, is likely to have some action on others, which would be reflected in what the website reports. They could still gain some points in a content analysis on indicators representing the other pillars.

Thus, information to assess the degree of sustainability implementation can be derived from publicly available information, such as websites and news. It may be somewhat tedious to carefully map all the information to the DJSI criteria, but it is not excessively difficult. The in-depth interviews here demonstrate that such publicly available information reflects internal operations fairly well in a general sense. As a research issue, then, content analysis of such information is sufficient for getting a reasonable overview of how well companies implement sustainability, which confirmed RQ2. Of course, inside interviews would still be needed for understanding various implementation details, as well as the corporate culture behind how companies operate in sustainable modes, and whether they have any particular focus on specific pillars.

The most important issue in this study is about how internal sustainability efforts directly relate to a favorable brand image among the companies' customers, which was RQ3. Generally, companies implementing sustainability more strongly have a stronger sustainability brand image. While not a perfect mapping, consumer perceptions of sustainable brands roughly match the companies' sustainability efforts. The slight discrepancy between DJSI scores and consumer perceptions is easy to explain by how well corporate communications are able to convey knowledge of sustainability initiatives. Consumers perceive sustainability efforts through the companies' products and CSR initiatives which are communicated publicly. They tend to build sustainable brand image from their own experience with the products, and from marketing and PR activities they see in various types of media.

Consumers are aware of three-pillar concepts, and prefer to see them implemented by the companies whose products they buy, although most are willing to accept that at the moment, the companies might not yet be doing all three. Previous literature has already (somewhat infrequently) suggested that even with a focus on, for example, green, consumers may well be concerned with aspects of the other pillars [13,97,98]. This is implied in the interconnectedness of the pillars [7], and a few consumers in our study already feel that three pillars are necessary for true sustainability. Many companies are already implementing the three pillars into their business practices and policies. Once they communicate this more effectively, consumers will not need to accept brand images that do not clearly deliver all three. Some recent work has already discussed how to implement three-pillar sustainability into product design [99], and a number of studies have confirmed the importance of communicating about various sustainability issues [90–92].

These brand image results reflect the current partial fragmentation of sustainability segments into individual pillars for many consumers. They tend to perceive companies to

be strong on environmentalism, or on CSR. (The economic pillar was not as strong in this study). This fragmentation, however, is likely to dissipate. Nearly all of the respondents here were aware of three-pillar sustainability and see interconnections among the pillars. A few already consider comprehensive sustainability rather than an individual pillar in their brand choice, and more expect the segments to gradually merge in the future. These issues all have implications both for research on conceptualizing how sustainability implementation works in the market, and for managerial practice.

### 4.1. Conceptual Implications

The first observation is that empirical research on sustainability as related to brand image has largely focused on a single pillar, usually the environmental or social one. Many companies, however, seem well aware of the need for three-pillar sustainability, and are implementing it. Industry, then, is somewhat ahead of academic research in sustainability thinking and action, if not necessarily in conceptualizing how this comprehensive sustainability works in the market. Conceptual discussions are available, but there is a shortage of empirical work to build an understanding of the more comprehensive view of sustainability beyond individual pillars [13]. For most large companies, the first step in filling this gap is relatively simple, since publicly available information seems sufficient to give a basic overview of what they do. Research can examine how well the degree of sustainability implementation correlates with brand image. Of course, many things contribute to the image, but sustainability should be one of the major factors in segments that value sustainability.

Here, the second key observation is that, just like companies, consumers are also aware of three-pillar sustainability. Some evidence has suggested that many consumers think of a single pillar, often the environmental one (e.g., [100]). Our data indicates that consumers do think of all three, even if many do not yet factor them all into their perceptions of brand image or product choice (which depends substantially on brand image). Some already do, but in general, we can still see specific pillar segments, such as consumers who value greenness or strong CSR. However, nearly all the respondents here already see the interconnections among the pillars. Many feel that the integration of the separate pillars into a more comprehensive whole is the long-term trend. An extension of the research just noted needs to examine how well three pillars are factored into companies' product design and marketing communications.

### 4.2. Managerial Implications

Sustainability is a strong component of a favorable brand image to many consumers. Until now, the preference among these sustainability-oriented consumers for 'sustainable brands' has largely focused on one pillar, usually either environmental or social. In other words, sustainability market segments were initially small and fragmented. It is well known now that they are growing, but in addition, the different pillar segments are beginning to merge. In the long run, companies will not be able to claim that they are sustainable by focusing mostly on the green, contributing to social development, or helping build local economies. 'Sustainable' to consumers is beginning to become a comprehensive three-pillar concept.

Figure 2 illustrates what is going on in the market. The three pillars started as small separate market niches. Companies could focus on one or the other, and do well in it (with actual good performance on the pillar), but did not necessarily have to worry about the other pillars. Many companies, however, are now aware of the long-term trend. The segments are growing, and they are merging. Eventually, a comprehensive sustainability segment will become the mainstream segment.

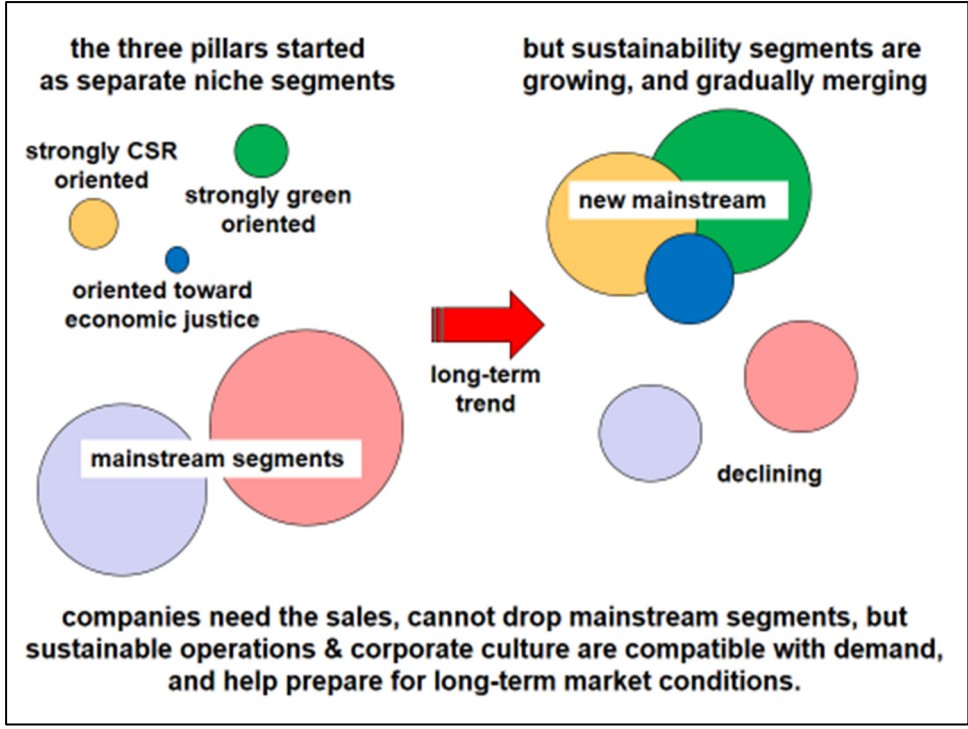

**Figure 2.** Evolution of sustainability segments. Source: authors.

Early in this process, the mainstream segments were not necessarily concerned with (any pillar of) sustainability. Some literature suggests that these consumers are not always willing to pay for green, or social (e.g., [101]). Most respondents in our interviews are willing to pay a premium because sustainable product characteristics and sustainable company operations have value to them. Some segments of the market, of course, remain not very sustainability-oriented, but there is little conflict except at the low end among strongly price-oriented consumers. Companies known for sustainability now are gaining the perception that they are concerned about high product quality. Consumers who value quality may not care about the greenness of the product or its social inclusiveness, but they may be willing to pay for the quality. Sustainability does not harm brand image in mainstream segments, even if it may not help it.

*4.3. Conclusions, Limitations, and Future Research*

One obvious limitation of this study is that it only examined large Thai companies which mainly sell household durables. As noted, some big companies do sell these products, but are not in the sample because the product category is a small part of their overall operations. They are likely to show similar patterns, but future research would need to confirm that companies in other industries approach sustainability similarly (or not). Further, some companies that are mainly in this product category do not yet do much about sustainability and were not examined here. Anecdotal evidence (from the authors' occasional executive seminars) suggests that they still mainly focus on the big mainstream segments in Figure 2, and have not assessed the long-term trends very thoroughly. Likely, they will need to catch up on their sustainability image eventually or be trapped in declining segments that may not support their current sales volume. Future research on how easy or difficult it is to play catch-up on these issues is needed.

In addition, some findings may not be applicable to small-scale companies. Public information for large companies is widely available on their websites and social media, and many consumers are aware of their brands. Small companies usually do not have very extensive websites or social media sites, so the content analysis used here to score companies on the DJSI might not work. It is also unknown whether the much more limited

workforce would allow implementation of comprehensive sustainability anyway, rather than requiring more focus on individual pillars. Most small companies would also have a smaller base of consumers who know the brand.

We did not explore marketing communications in much detail, but it was clear that the large companies have at least some, and often have been using it for many years. There were already indications that communications can have some impact on how aware consumers are of internal sustainability efforts. Future research needs to examine how marketing communications can support full three-pillar sustainability. In addition, small companies usually have much smaller communications budgets, so future research also needs to examine how small companies can use such communications if they opt for more comprehensive sustainability.

Finally, these findings might not represent other types of products well, especially for low-involvement goods where consumers do not think very much about the products. Consumer durables here are relatively high-involvement, where consumers think somewhat carefully about their product choices. This level of conscious assessment of product characteristics is rarely present for low-involvement products (e.g., [102,103]). Possible future research should investigate the impact of marketing communications on sustainable brand image in such low-involvement cases.

Clearly small-sample qualitative studies cannot demonstrate a phenomenon in general. However, this study does demonstrate that the assertions in the three RQ hold in this context. Companies are engaged in three-pillar sustainability, there are different levels of implementation, consumers use sustainability in constructing their brand image, and the market is moving toward a full three-pillar understanding of sustainability. In Siggeldow's [104] terminology, this qualitative research demonstrates that these three RQ are 'plausible'. Thus, the research points out paths that need to be explored as orientation toward sustainability continues to grow among companies and in markets.

**Author Contributions:** Conceptualization, P.C., M.S.; Methodology, P.C., M.S.; Software, not applicable. Validation, P.C., M.S.; Formal analysis, P.C., M.S.; Investigation, P.C.; Resources, P.C.; Data curation, P.C.; Writing—original draft preparation, P.C.; Writing—Review and editing, M.S.; Visualization, P.C., M.S.; Supervision, M.S.; Project administration, P.C.; Funding acquisition, not applicable. All authors have read and agreed to the published version of the manuscript.

**Funding:** This research received no external funding.

**Institutional Review Board Statement:** Institutional Review Board, Institute for Population and Social Research, Mahidol University.Protocol No.: IPSR-IRB-2022-153.

**Informed Consent Statement:** Informed consent was obtained from all subjects involved in the study.

**Data Availability Statement:** Much of the qualitative data is in the article in the form of quotes. The full master file cannot be shared without compromising promised anonymity.

**Conflicts of Interest:** The authors declare no conflict of interest.

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
