# Peer review of "Three-Pillar Sustainability and Brand Image: A Qualitative Investigation in Thailand’s Household Durables Industry"

_sustainability, doi:10.3390/su141811699_

Round 1
Reviewer 1 Report
Thanks a lot for this interesting paper. I recommend just a few minor revisions:
- before using the abbreviation "CSR" you should describe it
- RQ should start with the first letter big "How..."
- I think "literature discussion" is a nonsense
- after the section "Results" I miss the summary of results for the better orientation of the reader
- in the discussion I miss the same studies and their comparison
Author Response
in file

Reviewer 2 Report
While I acknowledge the author's efforts, I feel that the paper requires significant revision to warrant publication.
Introduction
The introduction requires a bit of work to get it strengthened. The main argument seems to focus on the lack of attention on the three pillars, and I feel that is not a solid case to warrant a study. For example, authors stated that ‘There is little work on whether most companies actually fully implement all three pillars, either internally, or externally in their brand images, and even some sustainability indices may lack full coverage’. Yet there is no evidence to support this statement. Second, there are quite a good number of papers and therefore ‘little work’ as used above is incorrect (see appendix A). Again, the flow of the introduction is problematic as it seems to be discussing so many things without a clear path. Please strengthen your arguments by explicitly pointing out what past studies have not done and why it needs to be done. Doing so will point out your theoretical contribution. Second, you need to also add the organisation of the rest of the study.
Literature review
The literature review requires significant improvement. The arguments presented are not strong and fails to show the need for your study.
Methods
The methods section requires significant improvement. There are several questions on the methodology which require clarity:
1. The authors claim they did an initial online search which revealed 9 companies; meanwhile the first author claims he/she has an indepth knowledge of the industry. Authors need to use other means to identify more companies. A brief about the industry and key participants will also help.
2. Did the authors also try other public databases?
3. How did you arrive at the 3 companies?
4. How did you select your participants? How did you arrive at the three?
5. Did you get their consent before interviewing them?
6. Did you get any ethical approval?
7. How did you assure them of anonymity?
8. How did you analyse your data?
These are essential questions your method section should address for us to see your findings as valid and reliable.
Findings
The weaknesses in the methods section cast doubts on the findings. For a topic such as this, with three participating organisations, I expect to see at least 5 interviewees per organisation or saturation. There are questions about the depth of your findings. However, keep in mind that qualitative research aims to achieve depth and richness and identify nuances that are not clear in your current manuscript.
Discussion, theoretical and managerial implications
While the author attempts to discuss the findings, I do not see a clear comparison to previous studies. As it stands, it’s unclear how similar or different the findings are. Remember to begin by discussing the research question and whether it was answered in the research paper based on the findings. Highlight any unexpected and exciting findings and link them to the research question. Point out some previous studies and draw comparisons on how your study is different.
Second, the theoretical implications of their findings need to be strengthened.
Appendix A
Author Response
in file

Reviewer 3 Report
This study examines the degree to which triple bottom line-like conceptualizations of sustainability are implemented in a particular industry in Thailand, and the degree to which these implementations are reflected in consumer images of the companies involved. I think the paper provides an interesting contribution by providing in-depth insights into synergies between the three pillars, and the studies seem well designed and conducted. I do have some comments that hopefully are useful to further strengthen the paper.
1. In the introduction, I think the study’s contribution could be more clearly articulated. Particularly, on page 2, the authors say that “research on consumer response is somewhat fragmented”. However, I don’t think the remainder of this paragraph really clarifies this point. In what sense exactly is previous research fragmented? I presume what the authors mean here is that previous studies have typically focused on only one or two of the ‘pillars’ (as they state in other places), but that is not quite clear to me here.
2. Throughout the results, the authors make regular comparisons between higher- and lower-scoring firms, which I think is good. However, this comparison seems to be missing on page 9 when the authors conclude that “continuous improvement is prominent in the high-score company”. Was such continuous improvement less apparent in lower-scoring companies?
3. In the discussion, the authors conclude that “content analysis of such information is sufficient for getting a reasonable overview of how well companies implement sustainability” (page 14). I agree that it is encouraging that the results of the content analysis and the interviews are reasonably consistent, but it seems there is also a possibility that the results of both methods would be biased in the same direction. Both are basically messages from the firms’ managers, who presumably have incentives to portray their firms in a positive light. I think this might be something to discuss as a potential limitation.
4. A few smaller points:
· On page 1, I presume that “progress toward incorporating… is somewhat mixed in the literature” should be either something like “evidence about incorporating… is somewhat mixed in the literature” or “progress toward incorporating… is somewhat lacking in the literature”. “Progress” and “mixed” do not quite seem to match with each other.
· Similarly, on the same page, “implement” and “in their brand images” do not quite seem to match with each other. Brand images are ‘created’ by consumers/stakeholders, not by brands, I think.
Author Response
in file

Round 2
Reviewer 2 Report
The authors did not address my concerns
Author Response
in file '01 response to editor revision2'
